# In Vitro Bioaccessibility and Antioxidant Activity of Coffee Silverskin Polyphenolic Extract and Characterization of Bioactive Compounds Using UHPLC-Q-Orbitrap HRMS

**DOI:** 10.3390/molecules25092132

**Published:** 2020-05-02

**Authors:** Luigi Castaldo, Alfonso Narváez, Luana Izzo, Giulia Graziani, Alberto Ritieni

**Affiliations:** 1Department of Pharmacy, University of Naples “Federico II”, Via Domenico Montesano 49, 80131 Napoli, Italy; luigi.castaldo2@unina.it (L.C.); alfonso.narvaezsimon@unina.it (A.N.); luana.izzo@unina.it (L.I.); giulia.graziani@unina.it (G.G.); 2Department of Clinical Medicine and Surgery, University of Naples “Federico II”, Via S. Pansini 5, 80131 Naples, Italy; 3Staff of UNESCO Chair on Health Education and Sustainable Development, Federico II University, 80131 Naples, Italy

**Keywords:** polyphenols, chlorogenic acids, coffee silverskin, bioaccessibility

## Abstract

Coffee silverskin (CS), the main by-product in the coffee industry, contains a vast number of human health-related compounds, which may justify its exploitation as a functional food ingredient. This study aimed to provide a comprehensive analysis of the polyphenolic and alkaloid profile through UHPLC-Q-Orbitrap HRMS analysis. The bioaccessibility of total phenolic compounds and changes in the antioxidant activity during an in vitro gastrointestinal digestion were also evaluated through spectrophotometric tests (TPC by Folin-Ciocalteu, ABTS, DPPH, and FRAP), to elucidate their efficacy for future applications in the nutraceutical industry. Caffeoylquinic and feruloylquinic acids were the most representative polyphenols, with a mean concentration of 5.93 and 4.25 mg/g, respectively. Results showed a high content of caffeine in the analyzed CS extracts, with a mean value of 31.2 mg/g, meaning a two-fold increase when compared to coffee brews. Our findings highlighted that both the bioaccessibility and antioxidant activity of CS polyphenols significantly increased in each in vitro gastrointestinal digestion stage. In addition, the colon stage might constitute the main biological site of action of these antioxidant compounds. These results suggest that in vivo, the dietary polyphenols from CS might be metabolized by human colonic microflora, generating metabolites with a greater antioxidant activity, increasing their well-known beneficial effects.

## 1. Introduction

Coffee is one of the most popular and appreciated beverages around the world due to its pleasant flavor and stimulating properties [1]. The coffee industry has steadily grown in the Western countries and represents one of the most highly traded commodities [2]. According to the latest data reported by the International Coffee Organization (ICO), global coffee production reached 168.9 million of 60 kg bags in the year 2019/2020 [3]. Consequently, huge amounts of waste and by-products are generated in the coffee industry, which are often poured into the environment, contributing to soil and water pollution [4].

Currently, by-product valorization is an innovative and eco-friendly way to contribute to a sustainable development, promoting the use of alternative sources for obtaining bioactive compounds with potential market value [5]. Natural high value ingredients recovered from agro-industrial by-products, such as antioxidants, may be an economically attractive solution, which can be used effectively in nutraceutical formulations, functional foods, dietary supplements, fortified foods, or for their health-promoting properties [6].

Coffee silverskin (CS) is a thin tegument of green coffee beans and constitutes the main by-product in the coffee industry [7]. Despite the efforts made by some research groups to find an alternative use for this coffee by-product, CS remains underutilized as a raw material for industrial processes [8]. Previous studies indicated that CS contains a wide range of active molecules that may affect human health, which include alkaloids, polyphenols, and other high molecular weight compounds generated in the late stages of the Maillard reaction during the roasting process, such as melanoidins [9]. These active molecules are well recognized as powerful compounds involved in the prevention of lifestyle-related diseases [10,11,12].

Among alkaloids, caffeine is considered the most active component in coffee, as well as CS samples [13]. Scientific evidence supports a protective role of caffeine against several chronic diseases, including type 2 diabetes mellitus and Parkinson’s disease [14], attributing many beneficial effects derived from coffee consumption to the polyphenolic fraction. Dietary polyphenols are secondary metabolites present in many plants, namely in tea, coffee, fruits, vegetables, and red wine, among others [15]. An ever-expanding amount of scientific data suggests that the habitual dietary intake of polyphenols can contribute to maintaining human health, preventing from a wide range of age-related pathologies [16]. Chlorogenic acids (CGAs) represent the major polyphenols found in CS samples. CGAs are plant-derived phytochemicals with antioxidant properties [17] formed by the esterification of cinnamic acids, namely caffeic, *p*-coumaric, and ferulic acids with a common skeleton of quinic acid [18], which give rise to caffeoylquinic (CQA), dicaffeoylquinic acid (diCQA), *p*-coumaroylquinic (*p*CoQA), feruloylquinic (FQA), caffeoyl-feruloylquinic (CFQA), and feruloyl-caffeoylquinic acids (FCQA) [19]. Caffeoylquinic, *p*-coumaroylquinic, and feruloylquinic acids are some of the most important polyphenols reported in CS, as well as coffee brews. As reported by several investigations, CGA shows high bioaccessibility in humans and a wide range of important biological activities such as antioxidant and anti-inflammatory properties, becoming good candidates for medicinal applications [20,21,22]. As regards melanoidins, recent studies have also reported significant antioxidant potential due to the presence of low molecular compounds, such as CGAs, incorporated through non-covalent bonds during roasting of coffee beans [23].

According to Tagliazzucchi et al. [24] bioaccessibility represents one of the most relevant factors concerning the beneficial activity of polyphenols. Bioaccessibility is defined as the amount of a compound present in the gastrointestinal tract that may be considered available for absorption [25]. The actions of both human digestive and gut microbial enzymes represent an essential mechanism for releasing native polyphenols from components of the food matrix. Previous studies have suggested that polyphenols play a role in modulation of the gut microflora and activity, which may have positive implications in maintaining human health [26]. Campos-Vega et al. [27] investigated the bioaccessibility and antioxidant capacity of spent coffee after gastrointestinal digestion and colonic fermentation. The authors concluded that gastrointestinal digestion significantly increased the antioxidant activity compared to their respective non-digested samples. Fernandez-Gomez et al. [28] evaluated the in vitro bioaccessibility of CGAs extracted from CS samples using a simulated gastrointestinal digestion protocol, excluding the colon stage in the analysis. However, it has been reported that in the large intestine, coffee melanoidins might be metabolized by gut microbiota enzymes generating antioxidant compounds linked to them. In this line, a recent study reported that, during in vitro gastrointestinal digestion, low molecular weight compounds were released from melanoidins in the colon stage, resulting in a higher antioxidant activity compared to their respective non-digested sample [29]. On the other hand, there is a lack of information regarding the bioaccessibility of total phenolic compounds of CS extracts and changes in the antioxidant activity displayed during each stage of the gastrointestinal digestion, including the colon fermentation phase.

As far as the analytical methods are concerned, the determination of both CGAs and alkaloids in vegetal matrices is mainly based on liquid chromatography (LC) coupled to mass spectrometry (MS) [30]. In the last decade, there have been improvements in the LC technique with the development of ultra-high performance LC (UHPLC), which has led to shorter analysis time, higher peak efficiency, and higher resolution [31,32]. Moreover, high resolution mass spectrometers (HRMS), such as the quadrupole orbital ion trap analyzer (Q-Orbitrap), have been used coupled to UHPLC for the determination of polyphenols and alkaloids profile in several food matrices including cocoa, tea, coffee, and coffee by-products [33,34,35]. This methodology stands as a powerful tool for the identification of natural products in plant extracts due to its high sensitivity and specificity, allowing precise quantification based on exact mass measurement.

Therefore, this study aimed to provide a comprehensive analysis of CGAs (*n* = 11) and alkaloids (*n* = 4) contained in four different polar coffee silverskin extracts obtained from *Coffea arabica* L. through ultra-high performance liquid chromatography coupled to a high resolution Orbitrap mass spectrometry (UHPLC-Q-Orbitrap HRMS). Besides, the bioaccessibility of total phenol compounds and changes in the antioxidant activity during an in vitro gastrointestinal digestion were also evaluated, in order to elucidate their efficacy for future applications in the nutraceutical industry.

## 2. Results

### 2.1. High Molecular Weight Melanoidins’ Content

The content in high molecular weight melanoidins (HMWM) obtained from four different CS samples by an ultrafiltration technique is summarized in Table 1. Total content of HMWM found in different types of CS extracts ranged from 173.8 to 234.9 mg/g, with a mean value of 204.6 mg/g for all samples. 

### 2.2. Identification of CGAs and Alkaloids in the Analyzed CS Extracts though UHPLC-Q-Orbitrap HRMS

Identification of individual CGAs and alkaloids was conducted through UHPLC-Q-Orbitrap HRMS. For the purpose of achieving a good separation of the studied analytes, three different gradient programs were tested. By starting with 0% of Phase B, the authors obtained satisfactory results in terms of separation and peak shape for all target analytes. Accurately, the authors selected a C18 column with a polar modified surface and a stable stationary phase under 100% aqueous condition, as reported by the manufacturer [36], compatible with the chosen gradient. A total of 15 compounds were identified from different samples of CS. The exact mass for the studied compounds including ion assignment, theoretical and measured mass (*m*/*z*), accuracy, and sensitivity are shown in Table 2.

Experiments were achieved both in negative (ESI−) and in positive (ESI+) ion mode. All CGAs exhibited better fragmentation patterns producing the deprotonated molecule [M−H]^−^, whereas the ionization in the positive mode generated the protonated molecule [M + H]^+^ for alkaloids. 

Full-scan HRMS data acquisition captured all sample data, enabling the identification of untargeted compounds and retrospective data analysis without the need to re-run samples. The confirmation was based on the accurate mass measurement, elemental composition assignment, and MS/MS spectrum interpretation (Appendix A). The identification of structural isomers caffeoylquinic acid (CQA, *m*/*z* 353.08780), *p*-coumaroylquinic acid (*p*CoQA, *m*/*z* 337.09289), dicaffeoylquinic acid (diCQA, *m*/*z* 515.11950), and feruloyl-caffeoylquinic and caffeoyl-feruloylquinic acids (FCQA and CFQA, *m*/*z* 529.13245) was achieved by comparison of their fragmentation pattern and the retention times previously reported in the literature [33,37]. Separation of all the investigated analytes was carried out in a total run time of 9 min. Due to insufficient chromatographic separation, 4- and 5-FQA were quantified together.

### 2.3. Quantification and Semi-Quantification of CGAs and Alkaloids in the Analyzed CS Extracts

The CGAs and alkaloids found in extracted concentration from the CS extracts were analyzed through the UHPLC-Q-Orbitrap HRMS method. The quantitative determination of target analytes was performed using calibration curves at eight concentration levels. We obtained regression coefficients >0.990. Semi-quantification of compounds that had no standard to generate a curve was based on a representative standard of the same group.

In the herein analyzed samples, eleven CGAs were identified and semi-quantified. Table 3 shows the results expressed as the mean value ± SD of the CGAs detected in different analyzed CS extracts. CQA isomers represented from 13.5 to 45.6% of total CGAs in a concentration range between 1.62 and 5.93 mg/g. Moreover, 4-CQA was found as the most commonly detected CQA ranging from 0.45 up to 2.65 mg/g. As far as FQAs were concerned, the levels of the three isomers represented from 24.8 to 69.0% of total CGAs and semi-quantified at a concentration range from 4.21 to 8.30 mg/g. On the other hand, *p*CoQA, mainly represented by 3- and 5-*p*CoQA, was detected at mean values of 1.17 and 0.84 mg/g for all samples, respectively. Regarding 3-*p*CoQA, the CS4 sample showed a ten-fold increase compared with other samples assayed. As shown in Table 3, among diCQA, 3,5-diCQA was the most commonly detected compound, ranging from 0.08 to 1.4 mg/g, with a mean value of 0.72 mg/g. On the other hand, isomers 4,5-CFQA and 3,4-FCQA were less relevant compounds, being only found in one and two samples, respectively, at low concentrations. 

Apart from those, some important alkaloids including trigonelline, theobromine, theophylline, and caffeine were quantified and semi-quantified in the assayed samples, as shown in Table 4. Caffeine was the most commonly detected alkaloid, ranging from 27.7 up to 34.3 mg/g, with a mean value of 31.2 mg/g for all samples. Among the alkaloids, trigonelline was detected in all samples with a mean of 20.8 mg/g (range from 16.3 to 23.7 mg/g). Theobromine and theophylline were semi-quantified in the lowest amount for all the analyzed samples with a mean value of 0.1 and 0.2 mg/g, respectively. 

### 2.4. In Vitro Bioaccessibility of Total Phenolic Compounds in the Analyzed CS Extracts

The bioaccessibility of total phenol compounds of the analyzed CS extracts was measured by using a simulated gastrointestinal digestion and colonic fermentation. Therefore, TPC by the Folin-Ciocalteu assay was evaluated during all stages of the gastrointestinal digestion and compared to non-digested samples in order to obtain an overview of their bioaccessibility. Table 5 shows the mean values (mg GAE/g) ± SD of TPC for all samples in each stage of the in vitro gastrointestinal digestion and the percentage of increase in TPC (not digested vs. digested stage). Digested samples showed significantly higher TPC values through all the gastrointestinal digestion stages when compared with the non-digested ones (*p*-value < 0.05). As shown in Table 5, the percentage of increase in TPC during gastrointestinal digestion was between 20.9 and 169.5%, with the highest value after the colon stage (considered as Pronase E stage + Viscozyme L stage).

### 2.5. Antioxidant Activity of the Polyphenolic Extract after In Vitro Gastrointestinal Digestion

The antioxidant activity of soluble fractions obtained during the gastrointestinal digestion (oral stage (OS), gastric stage (GS), duodenal stage (DS), Pronase E stage (PS), and Viscozyme stage (VS)) and non-digested extract was assessed and compared by three different procedures (ABTS, DPPH, and FRAP). Table 6 shows the results obtained (mean value and SD), expressed as mmol of Trolox equivalent per kg of extract.

Digested samples showed significantly higher antioxidant activity through all the gastrointestinal digestion stages when compared with the not digested ones (*p*-value < 0.05). The results are summarized in Table 6. The percentage of increase in the antioxidant activity during gastrointestinal digestion ranged from 19.4 to 162.6%, from 19.1 to 211.1%, and from −4.5 to 134.2% for ABTS, DPPH, and FRAP, respectively.

The highest percentage of increase in the antioxidant activity was found in the colon stage for all samples investigated. The ABTS, DPPH, and FRAP values measured during the gastrointestinal digestion against their corresponding total phenolic content values were correlated, except for the FRAP assay in the oral stage, as shown in Table 7.

## 3. Discussion

The aim of the present study was to provide useful information regarding the bioactive components present in CS material such as polyphenols, alkaloids, and melanoidins in order to add value to this coffee by-product. The polyphenolic profile of aqueous CS extracts was determined through UHPLC-HRMS with an Orbitrap mass analyzer. In detail, eleven CGAs and four alkaloids were identified and semi-quantified in four different CS extracts. Concerning the occurrence of CGAs in CS extracts, the levels found in assayed samples showed a two- to three-fold increase when compared to aqueous CS extract previously analyzed through the HPLC-MS/MS method by Nzekoue et al. [21], who reported a total concentration up to 5.44 mg/g. The monitored compounds were three of 11 CGAs (3,5-diCQA, 3-, and 5-CQA) revealed in the analyzed extracts, which explained the different results observed. CGAs detected in the extracts were mainly composed of CQAs and FQAs. The biological activity of these molecules has been related to specific functions involved in maintaining health status, like increasing plasma levels of glutathione, delayed glucose absorption, reducing the LDL oxidative susceptibility, and the inflammatory process, helping to prevent from several disorders [38,39,40]. Among alkaloids, caffeine was the most commonly detected compound, with a mean value of 31.2 mg/g. Our data revealed a two-fold increase in the mentioned alkaloid compared with regular coffee brews (ranging from 10.9 to 16.5 mg/g) [41]. Currently, caffeine represents the main ingredient in most energy drinks and energy bars. These products usually contain added synthetic caffeine, due to the significant price differences between natural sources of caffeine (coffee beans, tea leaves, and guarana, among others) and synthetic caffeine sources [42]. Caffeine-enriched extracts from CS, at a low purchase cost, could represent a natural ideal source to be used in these formulations. Melanoidins appear as one of the most relevant compounds found in CS samples. Several studies indicated that the melanoidins may exert a wide range of important biological properties including antioxidant and prebiotic activity [43]. It has been reported that the phenolic compounds found in coffee green beans can be linked with carbohydrates, dietary fiber, and proteins during the Maillard reaction, which contribute to the formation of coffee melanoidins [44]. According to Tores de la Cruz et al. [45], low molecular weight phenolic compounds, such as CGA, are incorporated in the melanoidin skeleton during the roasting process, forming a fiber–antioxidant complex. Our results showed a very high melanoidin content in aqueous extracts from CS, suggesting their potential use in the design of new healthy foods or as functional ingredients.

The bioaccessibility of total phenolic compounds and changes in the antioxidant activity during an in vitro gastrointestinal digestion were also evaluated. The digestion simulated in vitro was performed according to the INFOGEST method [46] until the duodenal stage, while the combined action of Pronase E and Viscozyme L was used to reproduce the action of microbiota occurring in the colon stage. On the other hand, our findings highlighted that both the bioaccessibility and antioxidant activity of CS polyphenols significantly increased after the colonic stage. These results suggested that the dietary polyphenols might be metabolized by human colonic microflora, generating metabolites with a greater antioxidant activity, resulting in an increase of their beneficial effects. Moreover, a good correlation existed among data obtained from the ABTS, DPPH, and FRAP tests and TPC measured during the gastrointestinal digestion, suggesting that these methods could provide some reliable information on the antioxidant properties of compounds released during in vitro digestion. Similar results have also been observed by Campos-Vega [27], who reported a higher bioaccessibility of phenolic compounds during the colon fermentation in spent coffee samples. Some epidemiology studies and meta-analysis showed an inverse association between coffee consumption and the incidence of colon cancer [47,48]. This outcome appeared to be attributable to the content of the melanoidins and CGAs in coffee brews, the same compounds found in analyzed samples [49]. Recently, several studies proposed the melanoidins as a type of soluble Maillardized dietary fiber that passes through the upper gastrointestinal tract without being absorbed and during the colon stage should be metabolized by gut microbiota, resulting in the release of phenolic compounds linked to them [50]. On the other hand, a previous study [44] reported that CS extract efficiently modulated gut microbiota composition; in particular, CS extract rich in melanoidins was able to induce preferential growth of bifidobacteria, rather than clostridia and *Bacteroides* spp. with well-known beneficial effects, suggesting that the consumption of CS melanoidin may exert a prebiotic effect. Results highlighted a high content of melanoidins, as well as CGAs and alkaloids, suggesting that these extracts may represent a suitable raw material for bioactive compounds to be used in nutraceutical formulations, functional foods, dietary supplements, fortified foods, or for their health-promoting properties.

## 4. Materials and Methods 

### 4.1. Reagents and Materials

A total of four different samples of coffee silverskin (CS) produced by roasting coffee beans (*Coffea arabica* L.) of different origins (Congo: CS1, Nicaragua: CS2, Uganda: CS3, and Brazil: CS4) were provided from Evra Srl, Basilicata region, Italy. All materials were dried at 60 °C using a laboratory oven until the sample moisture reached a level from 6.6% to 6.9% (*w*/*w*). After drying, all samples were milled into powder using a laboratory mill and stored at room temperature in a cool and dry place, away from direct light.

The standards of polyphenols (purity >98%), namely chlorogenic acid, 3,4-dicaffeoylquinic acid, and caffeine, were purchased from Sigma-Aldrich (Milan, Italy). For the antioxidant assays, gallic acid, 2,2’-azino-bis-3-ethylbenzthiazoline-6-sulphonic acid (ABTS), 6-hydroxy-2,5,7,8-tetramethylchromane-2-carboxylic acid (Trolox), potassium persulfate, 1,1-diphenyl-2-picrylhydrazyl (DPPH), 2,3,5-triphenyltetrazolio chloride (TPTZ), anhydrous ferric chloride, hydrochloric acid, and sodium acetate were purchased from Sigma-Aldrich (Milan, Italy). 

The enzymes used to simulate gastrointestinal digestion: α-amylase (1000–3000 U/mg solid) from human saliva, pepsin (≥2500 U/mg solid) from porcine gastric mucosa, pancreatin (8× USP) from porcine pancreas, bacterial protease from Streptomyces griseus (called also Pronase E, ≥3.5 U/mg solid), Viscozyme L, and also the chemicals calcium chloride dihydrate (CaCl_2_·2 H_2_O), sodium hydroxide (NaOH), potassium chloride (KCl), potassium thiocyanate (KCNS), monosodium phosphate (NaH_2_PO_4_), sodium sulphate (Na_2_SO_4_), sodium chloride (NaCl), and sodium bicarbonate (NaHCO_3_) were purchased from Sigma-Aldrich (Milan, Italy).

Methanol (MeOH) and water (LC-MS grade) were acquired from Carlo Erba reagents (Milan, Italy), whereas formic acid (98–100%) was purchased from Fluka (Milan, Italy).

### 4.2. Polyphenols Extraction

The polyphenols extraction procedure developed by Mesías et al. [51] was selected as a starting point and then slightly modified, as follows: 20 g of sample were suspended in 200 mL of water at 80 °C. The mixture was stirred using a horizontal shaker (KS130 Basic IKA, Argo Lab, Milan, Italy) for 30 min at 300× *g* and then centrifuged (X3R Heraeus Multifuge, Thermo Fisher Scientific LED GmbH, Kalkberg, Germany) at 4900× *g* at 24 °C for 10 min. The supernatant was recovered, and the pellet was re-extracted using the same procedure above reported. The pooled supernatants were freeze-dried and stored at −18 °C until analysis.

### 4.3. High Molecular Weight Melanoidins’ Content

Quantification of HMWM was carried out according to the procedure described by Rufian-Henares [52] with minor modifications. Briefly, four milliliters 4 mL of water extract (5 mg/mL) was subjected to ultrafiltration using an Amicon Ultra-4, regenerated cellulose 10 kDa (Millipore, Milan, Italy) at 7500× *g* for 70 min at 4 °C. The retentate was refilled with 4 mL of water and washed three times in order to separate the low molecular weight compounds from the retentate. The content of HMWM was determined by weighing the freeze dried retentate obtained after dialysis. The results were expressed as mg/g CS extracts.

### 4.4. Ultra-High Performance Liquid Chromatography and Orbitrap High Resolution Mass Spectrometry Analysis

Chromatographic analysis was performed through an UHPLC system (Thermo Fisher Scientific, Waltham, MA, USA) equipped with a degassing system, a Dionex Ultimate 3000, a Quaternary UHPLC pump with a pressure tolerance of 1250 bar, an autosampler device, and a thermostated (25 °C) Gemini 1.7 µm (50 mm × 2.1 mm, Phenomenex) column. The injection volume was of 5 μL. The eluent phases were: Phase A (water with 0.1% formic acid *v*/*v*) and Phase B (acetonitrile with 0.1% formic acid *v*/*v*). The separation gradient was applied as follows: initial 0% B for 1 min, increased to 95% B in 1 min. The gradient was held for 0.5 min at 95% B and linearly decreased to 75% B in 2.5 min and decreased again to 60% B in 1 min. Afterward, the gradient switched back to 0% B in 0.5 min and was held for 2.5 min for column re-equilibration. The flow rate was 0.4 mL/min. The autosampler was set at 10 °C. The UHPLC system was coupled to a Q-Exactive Orbitrap mass spectrometer (UHPLC, Thermo Fischer Scientific, Waltham, MA, USA) equipped with an electrospray (ESI) source simultaneously operating in fast negative/positive ion switching mode (Thermo Scientific, Bremen, Germany). The acquisitions were conducted by setting full MS/all ion fragmentation (AIF) mode that used a full MS scan (without higher-energy collisional dissociation HCD fragmentation), followed by an AIF scan (with fragmentation energy applied). Full MS/AIF experiments were carried out with the settings: microscans, 1; automatic gain control (AGC) target, 1e6; maximum injection time, 200 ms; mass resolution, 35,000 full width at half maximum (FWHM) at *m*/*z* 200 for full MS analysis; whereas AIF scan conditions were: microscans, 1; AGC target, 1e5; maximum injection time, 200 ms; mass resolution, 17,500 FWHM at *m*/*z* 200, HCD energy, at 10, 20, and 45. In both cases, the instrument was set to spray voltage 3.5 kV, capillary temperature 275 °C, sheath gas 45 (arbitrary units), auxiliary gas 10 (arbitrary units), *m*/*z* range 80–1200, data acquisition in profile mode. The accuracy and calibration of the Q Exactive Orbitrap LC-MS/MS were checked daily using a reference standard mixture provided by the manufacturer. The mass tolerance window was set to 5 ppm in both full scan MS and AIF modes. Data analysis and processing were performed using Xcalibur software v. 3.0.63 (Xcalibur, Thermo Fisher Scientific, Waltham, MA, USA).

### 4.5. In Vitro Simulated Gastrointestinal Digestion

The in vitro gastrointestinal digestion was performed according to the harmonized protocol recently developed in the COST action INFOGEST network [46]. The solutions used to simulate gastrointestinal digestion fluids (SSF: simulated salivary fluid, SGF: simulated gastric fluid, and SIF: simulated intestinal fluid) were prepared according to the procedure described by Minekus et al. [46] and shown in Table 8.

Briefly, five grams of aqueous extract were suspended in 3.5 mL of SSF. After 1 min of stirring, zero-point-five milliliters of α-amylase solution (made up in SSF, 75 U/mL), 25 µL of 0.3 M calcium chloride, and 975 µL water were added. Then, the pH of the mixture was adjusted to 7 with HCl 1 M, and the solution was incubated in a shaker bath at 100× *g* at 37 °C for 2 min. Afterward, 7.5 mL SGF, 1.6 mL pepsin solution (made up in SGF, 2000 U/mL), 5 μL of 0.3 M calcium chloride, and 695 µL of water were added and thoroughly mixed. The pH of the solution was adjusted to 3 with HCl 1 M, and the mixture was incubated in a shaker bath at 100× *g* at 37 °C for 2 h. 

Then, in order to simulate the intestinal condition, eleven milliliters SIF, 5 mL pancreatin solution (made up in SIF, 100 U/mL of trypsin activity), 40 μL of 0.3 M calcium chloride, and 2.5 mL bile salt solution (65 mg/mL) were added, and the volume was adjusted to 20 mL with water. Then, the mixture was thoroughly mixed, and the pH of the solution was adjusted to 7 with NaOH 1 M. The mixture was incubated in a shaker bath at 100× *g* at 37 °C for 2 h and then centrifuged at 4900× *g* at 37 °C for 10 min. The supernatant was collected, and the remaining pellet was treated in order to determine the polyphenols’ bioaccessibility and antioxidant activity in the colon stage according to the procedure described by Annunziata et al. [53]. Therefore, in order to simulate the actions of the gut microbiota in the colon stage, five milliliters Pronase E solution (1 mg/mL) were added, and the pH was adjusted to 8 with NaOH 1 M. The mixture was incubated in a shaker bath at 100× *g* at 37 °C for 1 h. Finally, the solution was centrifuged at 4900× *g* at 37 °C for 10 min. The supernatant was collected, and the remaining pellet was treated with 150 µL Viscozyme L. The mixture was adjusted to pH 4 and incubated in a shaker bath at 100× *g* at 37 °C for 16 h, then centrifuged at 4900× *g* at 24 °C for 10 min.

At each in vitro digestion stage (OP: oral phase, GP: gastric phase, IP: intestinal phase, PO: Pronase E phase, and VO: Viscozyme L phase) the supernatants were collected and freeze-dried. 

### 4.6. Determination of the Antioxidant Activity

The antioxidant activity of soluble fractions obtained during the gastrointestinal digestion (OP, GP, IP, PO, and VO) and non-digested extract was assessed and compared spectrophotometrically by three different procedures (ABTS, DPPH, and FRAP).

#### 4.6.1. Determination of 2,2’-Azino-bis-3-ethylbenzthiazoline-6-sulphonic Acid Free Radical-Scavenging Activity

The ABTS assay was conducted based on the method reported by Re et al. [54] with minor modifications. Briefly, forty-four microliters of 2.45 mM aqueous potassium persulfate were added to 2.5 mL of aqueous ABTS (7 mM), following 16 h of incubation at room temperature in the dark. The ABTS solution was diluted with ethanol until an absorbance value of 0.75 (± 0.02) at 734 nm to obtain an ABTS radical working solution. The analysis was performed by adding 100 μL of the sample to 1 mL of ABTS radical working solution. The absorbance was monitored after 3 min at 734 nm. Results were expressed as mmol Trolox equivalents (TE) per kg of sample extract.

#### 4.6.2. Determination of 2,2-Diphenyl-1-picrylhydrazyl Free Radical-Scavenging Activity

Determination of DPPH free radical-scavenging activity was carried out according to the procedure described by Brand-Williams et al. [55] with minor modifications. Briefly, methanolic DPPH (4 mg in 10 mL) was diluted with methanol to an absorbance value of 0.90 (± 0.02) at 517 nm to obtain a DPPH radical working solution. The analysis was performed by adding 200 μL of sample to 1 mL of DPPH radical working solution. The decrease in absorbance was monitored after 10 min at 517 nm. Results were expressed as mmol Trolox equivalents (TE) per kg of sample extract.

#### 4.6.3. Determination of Ferric Reducing/Antioxidant Power Assay

The FRAP assay was conducted based on the method reported by Rajurkar et al. [56] with slight adaptations. Briefly, the FRAP solution contained 1.25 mL of TPTZ (2,4,6-tris(2-pyridyl)-s-triazine) solution (10 mM) in HCl (40 mM), 1.25 mL of FeCl_3_ (20 mM) in water, and 12.5 mL of acetate buffer (0.3 M, pH 3.6). Then, one-hundred-fifty microliters of sample were added to 2.850 mL of FRAP solution. The absorbance was monitored after 4 min at 593 nm. Results were expressed as mmol Trolox equivalents (TE) per kg of extract.

### 4.7. Determination of Total Phenolic Content 

Total phenolic content (TPC) was measured using the Folin-Ciocalteu colorimetric method described previously by Singleton et al. [57] with slight modifications. In brief, zero-point-one-two-five milliliters of properly diluted extract were mixed with 0.125 mL of Folin-Ciocalteu reagent, 0.5 mL of deionized water, and then incubated at room temperature for 6 min. Afterward, one-point-two-five milliliters of a 7.5% sodium carbonate solution were added to the mixture and adjusted to 3 mL with deionized water. The absorbance was measured at 760 nm after 90 min of incubation at room temperature. The results were expressed as mg of gallic acid equivalents (GAE)/g of sample.

### 4.8. Statistics and Data Analysis

Values were expressed as the average values ± standard deviation (SD) of at least triplicate experiments. Statistical analysis of the results was performed using STATA 12 (STATACorp LP, College Station, TX, USA). The differences between average values were evaluated by using Tukey’s test at the level of significance *p* < 0.05. The correlation coefficients among means was determined using Pearson’s method.

## 5. Conclusions

In summary, the characterization of the main bioactive compounds including CGAs (*n* = 11) and alkaloids (*n* = 4) in four coffee silverskin samples through UHPLC-Q-Orbitrap spectrometry measurement was carried out. Notably, the caffeine content found in the analyzed CS extracts was similar to the concentration reported in coffee brews and roasted coffee. Results highlighted that CQAs and FQAs were the most common CGAs detected in the analyzed samples. 

Moreover, our results suggested that CS polyphenol bioaccessibility and antioxidant activity increased during gastrointestinal digestion, and the colon stage might constitute the main biological site of action of these antioxidant compounds, suggesting their potential health benefits and justifying their exploitation as a functional food ingredient. The results herein obtained could represent a starting point for further in-depth studies regarding compounds released during in vitro digestion stages and after microbiota fermentation in order to evaluate the potential health benefits.

## Figures and Tables

**Table 1 molecules-25-02132-t001:** Melanoidin content in different analyzed coffee silverskin (CS) extracts.

Samples	Melanoidins
	(mg/g) ± SD
CS1	194.4 ± 2.1
CS2	173.8 ± 1.3
CS3	215.4 ± 2.6
CS4	234.9 ± 3.1

**Table 2 molecules-25-02132-t002:** Chromatographic and spectrometric parameters including ion assignment, theoretical and measured mass (*m*/*z*), retention time, accuracy, and sensitivity for the investigated analytes (*n* = 15). FCQA, feruloyl-caffeoylquinic acid; *p*CoQA, *p*-coumaroylquinic acid; CFQA, caffeoyl-feruloylquinic.

Compounds	Chemical	Adduct	Theoretical	Measured	Accuracy	Retention Time	LOD	LOQ
	Formula		Mass (*m*/*z*)	Mass (*m*/*z*)	(Δ mg/kg)	(min)	(mg/kg)	(mg/kg)
Trigonelline	C_7_H_7_NO_2_	[M + H]^+^	138.05495	138.05487	−0.58	0.4	0.05	0.16
Theophylline	C_7_H_8_N_4_O_2_	[M + H]^+^	181.07200	181.07193	−0.39	3.63	0.05	0.16
Theobromine	C_7_H_8_N_4_O_2_	[M + H]^+^	181.07200	181.07204	0.22	3.74	0.05	0.16
5-CQA	C_16_H_18_O_9_	[M − H]^−^	353.08780	353.08780	0.00	3.81	0.05	0.14
Caffeine	C_8_H_10_N_4_O_2_	[M + H]^+^	195.08765	195.08757	−0.41	3.84	0.05	0.16
3,4-FCQA	C_26_H_26_O_12_	[M − H]^−^	529.13245	529.13245	0.00	3.85	0.05	0.16
4-CQA	C_16_H_18_O_9_	[M − H]^−^	353.08780	353.08768	−0.34	3.9	0.05	0.14
3-CQA	C_16_H_18_O_9_	[M − H]^−^	353.08780	353.08762	−0.51	3.94	0.05	0.14
3-*p*CoQA	C_16_H_18_O_8_	[M − H]^−^	337.09289	337.09232	−1.69	3.96	0.05	0.14
5-*p*CoQA	C_16_H_18_O_8_	[M − H]^−^	337.09289	337.09290	0.03	3.99	0.05	0.14
3-FQA	C_17_H_20_O_9_	[M − H]^−^	367.10346	367.10303	−1.17	3.96	0.05	0.14
4 + 5-FQA	C_17_H_20_O_9_	[M − H]^−^	367.10346	367.10309	−1.01	4.01	0.05	0.14
3,5-diCQA	C_25_H_24_O_12_	[M − H]^−^	515.11950	515.11993	0.83	4.11	0.05	0.16
4,5-CFQA	C_26_H_26_O_12_	[M − H]^−^	529.13245	529.13495	4.72	4.14	0.05	0.16
3,4-diCQA	C_25_H_24_O_12_	[M − H]^−^	515.11950	515.12103	2.97	4.22	0.05	0.16

**Table 3 molecules-25-02132-t003:** Chlorogenic acids’ (CGAs) content in the analyzed CS extracts. Results are shown as the mean value ± SD (*n* = 3).

Compounds	CS1	CS2	CS3	CS4
	Average (mg/g) ± SD	Average (mg/g) ± SD	Average (mg/g) ± SD	Average (mg/g) ± SD
3-CQA	0.21 ± 0.02	0.40 ± 0.04	0.51 ± 0.03	1.54 ± 0.06
4-CQA	0.45 ± 0.03	2.46 ± 0.12	2.65 ± 0.11	2.61 ± 0.04
5-CQA	0.96 ± 0.05	2.38 ± 0.61	0.85 ± 0.09	1.78 ± 0.06
3-*p*CoQA	0.04 ± 0.001	0.21 ± 0.04	0.15 ± 0.03	4.30 ± 0.31
5-*p*CoQA	0.31 ± 0.04	0.24 ± 0.08	0.13 ± 0.01	2.67 ± 0.04
3-FQA	1.51 ± 0.16	0.87 ± 0.09	0.75 ± 0.04	0.79 ± 0.05
4 + 5-FQA	6.79 ± 0.42	5.36 ± 0.19	3.46 ± 0.20	3.46 ± 0.09
3,4-diCQA	0.05 ± 0.01	n. d.	n. d.	n. d.
3,5-diCQA	1.40 ± 0.11	0.69 ± 0.08	0.08 ± 0.01	n. d.
3,4-FCQA	0.25 ± 0.06	n. d.	0.19 ± 0.03	n. d.
4,5-CFQA	0.07 ± 0.01	n. d.	n. d.	n. d.
**Total**	12.04	12.61	8.77	17.15

The differences between average values were evaluated by using Tukey’s test at the level of significance *p* < 0.05.

**Table 4 molecules-25-02132-t004:** Alkaloids’ content in the analyzed CS extracts. Results are shown as the mean value ± SD (*n* = 3).

Alkaloids	CS1	CS2	CS3	CS4
	Average (mg/g) ± SD	Average (mg/g) ± SD	Average (mg/kg) ± SD	Average (mg/g) ± SD
Trigonelline	22.5 ± 0.3	16.3 ± 0.2	20.4 ± 0.3	23.7 ± 0.3
Theobromine	0.3 ± 0.01	0.2 ± 0.02	0.3 ± 0.03	0.0 ± 0.01
Theophylline	0.1 ± 0.02	0.1 ± 0.01	0.1 ± 0.02	0.1 ± 0.01
Caffeine	33.6 ± 0.4	29.2 ± 0.5	27.7 ± 0.4	34.3 ± 0.6
**Total**	56.39	45.73	48.47	58.19

The differences between average values were evaluated by using Tukey’s test at the level of significance *p* < 0.05.

**Table 5 molecules-25-02132-t005:** Bioaccessibility of total phenolic content evaluated by the Folin-Ciocalteu method in non-digested samples and during the simulated in vitro digestion.

Samples	TPC in CS1		TPC in CS2		TPC in CS3		TPC in CS4	
	mg GAE/g ± SD	%	mg GAE/g ± SD	%	mg GAE/g ± SD	%	mg GAE/g ± SD	%
Not digested	5.7 ± 0.1		5.8 ± 0.2		6.5 ± 0.2		6.9 ± 0.3	
Digestion Stage								
OS	7.4 ± 0.3	28.2	7.3 ± 0.3	24.7	8.5 ± 0.3	33.0	8.3 ± 0.3	20.9
GS	7.6 ± 0.3	32.0	7.8 ± 0.4	34.5	10.1 ± 0.5	59.1	9.4 ± 0.6	37.0
DS	8.4 ± 0.4	46.7	8.7 ± 0.4	48.8	10.3 ± 0.4	62.2	10.1 ± 0.2	47.2
PS	8.1 ± 0.5	40.9	8.3 ± 0.5	42.7	9.7 ± 0.3	52.7	10.4 ± 0.4	51.5
VS	6.2 ± 0.5	8.0	6.9 ± 0.6	18.6	7.4 ± 0.4	16.2	8.1 ± 0.3	18.0
Total colon stage	14.3 ± 0.5	148.9	15.2 ± 0.6	161.3	17.1 ± 0.4	168.9	18.5 ± 0.4	169.5

OS: oral stage; GS: gastric stage; DS: duodenal stage; PS: Pronase E stage; VS: Viscozyme stage; GAE: gallic acid equivalents; TPC: total phenolic content; %: percentage of increase or decrease in TPC (not digested vs. digested stage). The differences between average values were evaluated by using Tukey’s test at the level of significance *p* < 0.05.

**Table 6 molecules-25-02132-t006:** Antioxidant activity of digested samples evaluated by ABTS, DPPH, and FRAP assays expressed as mmol Trolox/kg of CS extract.

	ABTS mmol Trolox/kg ± SD				DPPH mmol Trolox/kg ± SD				FRAP mmol Trolox/kg ± SD			
	CS1	CS2	CS3	CS4	CS1	CS2	CS3	CS4	CS1	CS2	CS3	CS4
Not digested	23.5 ± 0.2	19.1 ± 0.3	28.3 ± 0.3	29.1 ± 0.4	8.1 ± 0.1	6.7 ± 0.1	9.5 ± 0.2	11.2 ± 0.2	14.4 ± 0.3	15.8 ± 0.3	18.3 ± 0.5	17.2 ± 0.4
Digestion Stage												
OS	30.5 ± 0.3	28.7 ± 0.4	33.8 ± 0.4	37.0 ± 0.5	10.3 ± 0.2	10.0 ± 0.1	12.0 ± 0.1	13.6 ± 0.2	17.2 ± 0.1	19.2 ± 0.2	22.9 ± 0.3	21.8 ± 0.4
GS	30.8 ± 0.3	24.7 ± 0.3	36.8 ± 0.4	36.9 ± 0.5	9.7 ± 0.2	8.1 ± 0.1	14.0 ± 0.3	13.5 ± 0.4	16.2 ± 0.4	18.7 ± 0.3	17.5 ± 0.5	24.0 ± 0.6
DS	43.4 ± 1.6	35.3 ± 0.8	49.4 ± 1.1	52.4 ± 0.6	20.5 ± 0.2	10.7 ± 0.3	22.9 ± 0.3	22.8 ± 0.4	16.5 ± 0.5	16.7 ± 0.2	19.3 ± 0.3	22.2 ± 0.3
PS	36.0 ± 0.4	22.1 ± 0.2	32.7 ± 0.4	42.6 ± 0.8	16.5 ± 0.1	13.0 ± 0.3	18.5 ± 0.2	21.2 ± 0.3	17.5 ± 0.1	16.6 ± 0.3	19.7 ± 0.5	21.6 ± 0.4
VS	25.7 ± 0.7	20.4 ± 0.3	29.3 ± 0.4	31.9 ± 0.5	8.7 ± 0.1	7.5 ± 0.1	10.5 ± 0.3	12.2 ± 0.3	11.7 ± 0.1	10.6 ± 0.2	13.7 ± 0.1	18.6 ± 0.2
Total colon stage	61.7 ± 0.3	42.6 ± 0.2	61.9 ± 0.4	74.5 ± 0.7	25.2 ± 0.1	20.5 ± 0.2	29.0 ± 0.3	33.5 ± 0.3	29.2 ± 0.1	27.2 ± 0.3	27.2 ± 0.3	40.3 ± 0.3

OS: oral stage; GS: gastric stage; DS: duodenal stage; PS: Pronase E stage; VS: Viscozyme stage; ABTS and DPPH: free-radical scavenging; FRAP ferric reducing/antioxidant power; Trolox equivalent antioxidant capacity (TEAC). The differences between average values were evaluated by using Tukey’s test at the level of significance *p* < 0.05.

**Table 7 molecules-25-02132-t007:** Correlation between TPC evaluated by Folin-Ciocalteu and antioxidant activity evaluated by the ABTS, DPPH, and FRAP methods. The correlation coefficients among means were determined using Pearson’s method.

Assay	Oral Stage *R*^2^	Gastric Stage *R*^2^	Duodenal Stage *R*^2^	Pronase E Stage *R*^2^	Viscozyme L Stage *R*^2^
ABTS	0.885	0.84	0.823	0.631	0.677
DPPH	0.882	0.941	0.652	0.875	0.82
FRAP	0.557	0.379	0.861	0.968	0.861

**Table 8 molecules-25-02132-t008:** Composition of stock solutions of simulated digestion fluid. SSF: simulated salivary fluid, SGF: simulated gastric fluid, and SIF: simulated intestinal fluid.

		SSF (pH 7)		SGF (pH 3)		SIF (pH 7)	
Salt Solution	Stock Concentration (mol/L)	mL of Stock Added to Prepare 0.4 L (mL)	Final Salt Concentration in Sample (mmol/L)	mL of Stock Added to Prepare 0.4 L (mL)	Final Salt Concentration in Sample (mmol/L)	mL of Stock Added to Prepare 0.4 L (mL)	Final Salt Concentration in Sample (mmol/L)
KCl	0.5	15.1	15.1	6.9	6.9	6.8	6.8
KH_2_PO_4_	0.5	3.7	1.35	0.9	0.9	0.8	0.8
NaHCO_3_	1	6.8	13.68	12.5	25	42.5	85
NaCl	2	-	-	11.8	47.2	9.6	38.4
MgCl_2_(H_2_O)_6_	0.15	0.5	0.15	0.4	0.12	1.1	0.33
NH_4_(CO_3_)_2_	0.5	0.06	0.06	0.5	0.5	-	-
CaCl_2_(H_2_O)_2_	0.3	-	1.5	-	0.15	-	0.6

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
