# Peer review of "In Vitro Bioaccessibility and Antioxidant Activity of Coffee Silverskin Polyphenolic Extract and Characterization of Bioactive Compounds Using UHPLC-Q-Orbitrap HRMS"

_molecules, 2020, doi:10.3390/molecules25092132_

Round 1

Reviewer 1 Report

The manuscript entitled "In Vitro Bioaccessibility and Antioxidant Activity of Coffee Silverskin Polyphenolic Extract and Characterization of bioactive compounds using UHPLC-Q-Orbitrap HRMS" presents a comprehensive analysis of the alkaloid and polyphenolic profile through UHPLC ‐ Q ‐ Orbitrap HRMS. The authors additionally undertook an analysis of the bioaccessibility of total phenolic compounds and changes in the antioxidant activity during an in vitro gastrointestinal digestion.

In my opinion, this type of research is extremely valuable, especially since there are many sources of useful biologically active compounds in nature that are lost in technological processes. Coffee silverskin is of interest to researchers because many data show that the compounds it contains can help to fight against chronic diseases caused by oxidation and inflammation.

In addition, coffee melanoidins possess various biological properties such as anticarcinogenic, anticariogenic, anti-inflammatory, antimicrobial, antioxidant or antihypertensive. The results described in the manuscript are very interesting and promising.

The main advantages of work are:

  1. Characteristics of the main biologically active compounds of coffee silverskin.
  2. Demonstration that the caffeine content found in CS extracts is similar to concentration reported in coffee brews and roasted coffee.
  3. Proof that CS polyphenol bioaccessibility and antioxidant activity increase during gastrointestinal digestion, and colon stage might constitute the main biological site of action of these antioxidant compounds, suggesting their potential health benefits and justifying their exploitation as a functional food ingredient.

In my opinion, the obtained results are a very good starting point for further in vivo analyzes, such as e.g. metabolic, pathological, or inflammatory parameters and gut microbiota composition monitoring.

I think the authors should emphasize the positive impact of biologically active compounds contained in coffee silverskin in the prevention of lifestyle diseases.

Author's abbreviated name should be added to Coffea arabica

Author Response

Response to reviewer 1

Manuscript ID: molecules- 788702
Type of manuscript: Article
Title: In Vitro Bioaccessibility and Antioxidant Activity of Coffee Silverskin Polyphenolic Extract and Characterization of bioactive compounds using UHPLC-Q-Orbitrap HRMS

Reviewer 1

The manuscript entitled "In Vitro Bioaccessibility and Antioxidant Activity of Coffee Silverskin Polyphenolic Extract and Characterization of bioactive compounds using UHPLC-Q-Orbitrap HRMS" presents a comprehensive analysis of the alkaloid and polyphenolic profile through UHPLC ‐ Q ‐ Orbitrap HRMS. The authors additionally undertook an analysis of the bioaccessibility of total phenolic compounds and changes in the antioxidant activity during an in vitro gastrointestinal digestion.

In my opinion, this type of research is extremely valuable, especially since there are many sources of useful biologically active compounds in nature that are lost in technological processes. Coffee silverskin is of interest to researchers because many data show that the compounds it contains can help to fight against chronic diseases caused by oxidation and inflammation.

In addition, coffee melanoidins possess various biological properties such as anticarcinogenic, anticariogenic, anti-inflammatory, antimicrobial, antioxidant or antihypertensive. The results described in the manuscript are very interesting and promising.

The main advantages of work are:

  1. Characteristics of the main biologically active compounds of coffee silverskin.
  2. Demonstration that the caffeine content found in CS extracts is similar to concentration reported in coffee brews and roasted coffee.
  3. Proof that CS polyphenol bioaccessibility and antioxidant activity increase during gastrointestinal digestion, and colon stage might constitute the main biological site of action of these antioxidant compounds, suggesting their potential health benefits and justifying their exploitation as a functional food ingredient.

In my opinion, the obtained results are a very good starting point for further in vivo analyzes, such as e.g. metabolic, pathological, or inflammatory parameters and gut microbiota composition monitoring.

  1. a) I think the authors should emphasize the positive impact of biologically active compounds contained in coffee silverskin in the prevention of lifestyle diseases.

- As suggested by reviewer 1,  the authors added this sentence in the manuscript: “These active molecules are well recognized as powerful compounds involved in the prevention of lifestyle-related diseases.”

  1. b) Author's abbreviated name should be added to Coffea arabica

- As suggested by reviewer 1,  the authors added the missing information as “Coffea arabica L.”

The authors thank reviewer for evaluating the manuscript.

Reviewer 2 Report

This is a remarkably interesting manuscript dealing with the revalorization of a coffee by-product (the coffee silverskin) due to its potential antioxidant capacity and the presence of bioactive substances with human health benefits. The manuscript is very well written, and the experiments well performed and discussed. The obtained results are very promising and very interesting for the scientific community dealing with the analysis and identification of natural products. In my opinion, this manuscript is suitable for publication in Molecules after major revision.

Introduction: The introduction is very well referenced, and it presents the topic correctly as for coffee properties, revalorization of coffee by-products, and bioavailability of potential bioactive substances. However, it lacks information regarding previously published analytical methodologies for the determination of, for instance, polyphenols in coffee samples, especially by employing UHPLC-HRMS techniques like the one used by the authors. In my opinion, this needs to be also addressed in the introduction to provide a wider range of information to the readers.

Line 113: [M-H]- is not the molecular ion. It is the deprotonated molecule. Please change “…producing the molecular ion [M-H]-,…” by “…producing the deprotonated molecule [M+H]-,…”

Lines 118- 122: Authors identified structural isomers by comparison of its fragmentation pattern and the retention times previously reported in the literature. Please, provide the references employed.

Table 2. In mass spectrometry [M+H]+ and [M-H]- are not adducts. They are the protonated and the deprotonated molecule, respectively. An adduct ion is an ion formed by adduction of ionic or neutral species to a molecule, but not the protonation or deprotonation of the molecule. I suggest changing the label “Adduct ion” by “Adduct”. Please, also correct in the table the assignment of the protonated molecules ([M+H]+ by [M+H]+). Also correct the label “Retention Time” by “Retention Time”. Also rewrite the table title: I suggest removing the term “optimized (most of these are experimental results not obtained from optimization, for example, the m/z values), and change “adduct ion” by “ion assignment”.

Line 132: How polyphenols were quantified? According to the reagents and materials section only standards for chlorogenic acid, 3,4-dicaffeoylquinic acid, and gallic acid were obtained. If authors have not employed the pure standard of each compound for its quantitation, this needs to be explained in the manuscript, and they cannot talk about quantitation but semi-quantitation, because when quantifying an analyte by employing the standard of another one (even if it is family-related) there is never enough evidence that the response factor for both compounds in UHPLC-HRMS will be the same. Therefore, it is better to refer to a semi-quantitative approach.

Table 3: In the table title, I suggest removing (n=11) because expressed it that way it seems that 11 CS extracts were analyzed when authors mean the number of compounds. In contrast, I suggest indicating the number of replicates performed in the analysis (Results are shown as mean value ± SD (n=?)). Same comment for Table 4.

Line 164: Change “mean values (mg GAE/g±SD)” by “mean values (mg GAE/g) ± SD”. Standard deviation never is included in the mean value.

Table 5. The term TPC is not appearing in the Table, although it is defined at the of the table. Please, label better the term “%” in the table. Is it TPC %?

Lines 180-181: Please define OP, GP, IP, PO, and VO, or at least include these abbreviations in the tables.

Line 209: Change “UHPLC Orbitrap mass analyzer” by “UHPLC-HRMS with an Orbitrap mass analyzer”.

Line 220: “two-to-fold”: Do you mean two-fold? Or two-to-?-fold?

Lines 287-295: Do the authors have any idea of the recoveries? This will again affect the quantitation results. If recoveries are not known, the authors cannot refer to concentrations in the CS extracts, but “extracted concentration form the CS extracts”. They cannot be sure if the extraction was 100% efficient.

Line 312: The UHPLC pump is not working at 1250 bar. This is the highest pressure that the UHPLC pump supports. The working pressure will depend on the column and the chromatographic conditions (mobile phase components and flow rate).

Line 312-313. The authors need to provide the stationary phase of the employed Gemini 1.7 313 µm (50 mm × 2.1 mm, Phenomenex) column. Is it a C18 column? In that case, it is a bit strange why the gradient begins with 100% water content. In fact, the gradient employed is quite strange if a C18 column is used, as the authors begin with 100% water, then increase very fast to 95% acetonitrile? How this gradient was optimized? Is it based on a previously reported work? I believe that some explanation needs to be included in the manuscript regarding the UHPLC-HRMS method employed.

Line 325: Please define FWHM (full width at half maximum)

Supplementary materials: Please add the corresponding label in the y-axis of all the spectra (Relative abundance, %).

Author Response

Response to reviewer 2

Manuscript ID: molecules- 788702
Type of manuscript: Article
Title: In Vitro Bioaccessibility and Antioxidant Activity of Coffee Silverskin Polyphenolic Extract and Characterization of bioactive compounds using UHPLC-Q-Orbitrap HRMS

Reviewer 2

Comments and Suggestions for Authors

This is a remarkably interesting manuscript dealing with the revalorization of a coffee by-product (the coffee silverskin) due to its potential antioxidant capacity and the presence of bioactive substances with human health benefits. The manuscript is very well written, and the experiments well performed and discussed. The obtained results are very promising and very interesting for the scientific community dealing with the analysis and identification of natural products. In my opinion, this manuscript is suitable for publication in Molecules after major revision.

a) Introduction: The introduction is very well referenced, and it presents the topic correctly as for coffee properties, revalorization of coffee by-products, and bioavailability of potential bioactive substances. However, it lacks information regarding previously published analytical methodologies for the determination of, for instance, polyphenols in coffee samples, especially by employing UHPLC-HRMS techniques like the one used by the authors. In my opinion, this needs to be also addressed in the introduction to provide a wider range of information to the readers.

- As suggested by reviewer 1, the authors added in the text details about published analytical methodologies for the determination of polyphenols in coffee samples, especially by employing UHPLC-HRMS techniques as: “As far as the analytical methods are concerned, the determination of both CGAs and alkaloids in vegetal matrices are mainly based on liquid chromatography (LC) coupled to mass spectrometry (MS) [27]. In the last decade, there have been improvements in the LC technique with the development of ultra-high performance LC (UHPLC), which has led to shorter analysis time, higher peak efficiency, and higher resolution [28-29]. Moreover, high-resolution mass spectrometers (HRMS), such as Quadrupole orbital ion trap analyzer (Q-Orbitrap), have been used coupled to UHPLC for the determination of polyphenols and alkaloids profile in several food matrices including cocoa, tea, coffee and coffee by-products [30-32]. This methodology stands as a powerful tool for the identification of natural products in plant extracts due to its high sensitivity and specificity, allowing precise quantification based on exact mass measurement”

b) Line 113: [M-H]-is not the molecular ion. It is the deprotonated molecule. Please change “…producing the molecular ion [M-H]-,…” by “…producing the deprotonated molecule [M+H]-,…”

- As rightly suggested by the reviewer 2, the authors changed the term “molecular ion” to “deprotonated molecule”

c) Lines 118- 122: Authors identified structural isomers by comparison of its fragmentation pattern and the retention times previously reported in the literature. Please, provide the references employed.

-As suggested by reviewer 4, the authors added the reference.

d) Table 2. In mass spectrometry [M+H]+and [M-H]-are not adducts. They are the protonated and the deprotonated molecule, respectively. An adduct ion is an ion formed by adduction of ionic or neutral species to a molecule, but not the protonation or deprotonation of the molecule. I suggest changing the label “Adduct ion” by “Adduct”. Please, also correct in the table the assignment of the protonated molecules ([M+H]+ by [M+H]+). Also correct the label “Retention Time” by “Retention Time”. Also rewrite the table title: I suggest removing the term “optimized (most of these are experimental results not obtained from optimization, for example, the m/z values), and change “adduct ion” by “ion assignment”.

-As suggested by the reviewer 2, the authors changed the term “Adduct ion” to “Adduct” and “Adduct ion” to “ion assignment”. Moreover, the authors corrected the label “Retention Tiome” to “Retention Time”. In addition, the authors removed the term “optimized”, and correct the molecules ([M+H]+ to [M+H]+.

e) Line 132: How polyphenols were quantified? According to the reagents and materials section only standards for chlorogenic acid, 3,4-dicaffeoylquinic acid, and gallic acid were obtained. If authors have not employed the pure standard of each compound for its quantitation, this needs to be explained in the manuscript, and they cannot talk about quantitation but semi-quantitation, because when quantifying an analyte by employing the standard of another one (even if it is family-related) there is never enough evidence that the response factor for both compounds in UHPLC-HRMS will be the same. Therefore, it is better to refer to a semi-quantitative approach.

-As rightly suggested by reviewer 2, the authors changed the term “quantitation” to “semi-quantitation”. The authors added in the text details as: “Quantitative determination of analytes was performed using calibration curves at eight concentration levels. We obtained regression coefficients >0.990. Semi-quantification of compounds that had no standard to generate a curve was based on a representative standard of the same group”.

f) Table 3: In the table title, I suggest removing (n=11) because expressed it that way it seems that 11 CS extracts were analyzed when authors mean the number of compounds. In contrast, I suggest indicating the number of replicates performed in the analysis (Results are shown as mean value ± SD (n=?)). Same comment for Table 4.

-As suggested by reviewer 2, in the table title, the authors removed the number of the analyzed CS samples and added the number of replicates performed in the analysis.

g) Line 164: Change “mean values (mg GAE/g±SD)” by “mean values (mg GAE/g) ± SD”. Standard deviation never is included in the mean value.

-As suggested by reviewer 2, the authors changed the term “(mg GAE/g±SD” to “(mg GAE/g) ± SD”.

h) Table 5. The term TPC is not appearing in the Table, although it is defined at the of the table. Please, label better the term “%” in the table. Is it TPC %?

-As suggested by reviewer 2, the authors added the term “TPC” in Table 5. Also, the authors clarify the meaning of the term “%” as “percentage of increase or decrease in TPC (not digested vs. digested stage)”.

i) Lines 180-181: Please define OP, GP, IP, PO, and VO, or at least include these abbreviations in the tables.

-As suggested by reviewer 2, the authors defined and included the abbreviations in the tables. In addition, the authors changed the abbreviations “OP, GP, IP, PO, and VO” to “OS, GS, DS, PS, and VS”.

J) Line 209: Change “UHPLC Orbitrap mass analyzer” by “UHPLC-HRMS with an Orbitrap mass analyzer”.

-As suggested by reviewer 2, the authors changed the term “UHPLC Orbitrap mass analyzer” to “UHPLC-HRMS with an Orbitrap mass analyzer”.

k) Line 220: “two-to-fold”: Do you mean two-fold? Or two-to-?-fold?

-As suggested by reviewer 2, the authors changed “two-to-fold” to “two-fold”.

l) Lines 287-295: Do the authors have any idea of the recoveries? This will again affect the quantitation results. If recoveries are not known, the authors cannot refer to concentrations in the CS extracts, but “extracted concentration form the CS extracts”. They cannot be sure if the extraction was 100% efficient.

-As suggested by reviewer 2, the authors refer the concentrations in the CS extracts as “extracted concentration form the CS extracts” and as “analyzed CS extracts”.

m) Line 312: The UHPLC pump is not working at 1250 bar. This is the highest pressure that the UHPLC pump supports. The working pressure will depend on the column and the chromatographic conditions (mobile phase components and flow rate).

- As suggested by reviewer 2,  the authors changed this sentence in order to clarify meaning as  “Quaternary UHPLC pump with pressure tolerance of 1250 bar”

  1. n) Line 312-313. The authors need to provide the stationary phase of the employed Gemini 1.7 313 µm (50 mm × 2.1 mm, Phenomenex) column. Is it a C18 column? In that case, it is a bit strange why the gradient begins with 100% water content. In fact, the gradient employed is quite strange if a C18 column is used, as the authors begin with 100% water, then increase very fast to 95% acetonitrile? How this gradient was optimized? Is it based on a previously reported work? I believe that some explanation needs to be included in the manuscript regarding the UHPLC-HRMS method employed.
  2. -As suggested by reviewer 2,  the authors added the missing information in the manuscript as: For the purpose of achieving a good separation of the studied analytes, three different gradient programs were tested. By starting with 0% of phase B the authors have obtained satisfactory results in terms of separation and peak shape for all target analytes. Accurately, the authors have selected a C18 column with a polar modified surface, a stable stationary phase under 100% aqueous condition as reported by the manufacturer [34], compatible with the chosen gradient.
  3. o) Line 325: Please define FWHM (full width at half maximum)

- As suggested by reviewer 2,  the authors defined the term “FWHM”

p) Supplementary materials: Please add the corresponding label in the y-axis of all the spectra (Relative abundance, %).

- As suggested by reviewer 2,  the authors added the corresponding label in the y-axis of all the spectra (Relative abundance, %).

The authors thank reviewer for evaluating the manuscript.

Round 2

Reviewer 2 Report

The authors followed all my suggestions to improve the manuscript quality. The manuscript can be accepted for publication.